# Navigating vaccination choices: The intersecting dynamics of institutional trust, belonging and message perception among Congolese migrants in London, UK (a reflexive thematic analysis)

**Alison F. Crawshaw**[1], **Tushna Vandrevala**[2], **Felicity Knights**[1], **Anna Deal**[1,3], **Laura Muzinga Lutumba**[4], **Sarah Nkembi**[4], **Lusau Mimi Kitoko**[4], **Caroline Hickey**[5], **Alice S. Forster**[6], **Sally Hargreaves**[1]*

1 Institute for Infection and Immunity, The Migrant Health Research Group, St George's, University of London, Cranmer Terrace, London, United Kingdom, 2 Faculty of Health, Science, Social Care and Education, Centre for Applied Health and Social Care Research, Kingston University London, London, United Kingdom, 3 Department of Public Health and Policy, London School of Hygiene and Tropical Medicine, London, United Kingdom, 4 Hackney Congolese Women Support Group, c/o Hackney CVS, The Adiaha Antigha Centre, London, United Kingdom, 5 Hackney Refugee and Migrant Forum and Hackney CVS, The Adiaha Antigha Centre, London, United Kingdom, 6 Our Future Health, New Bailey, Manchester, United Kingdom

* s.hargreaves@sgul.ac.uk

## Abstract

The COVID-19 pandemic disproportionately impacted intersectionally marginalised migrants, revealing systemic disparities in health outcomes and vaccine uptake. Understanding the underlying social and structural factors influencing health behaviours is necessary to develop tailored interventions for migrants, but these factors have been seldom explored. This qualitative study aimed to explore contextual factors shaping COVID-19 vaccination decision-making among Congolese migrants in the UK. A community-based participatory research study was designed and led by a community-academic partnership in London, UK (2021–2022). Peer-led, semi-structured interviews were conducted in Lingala with 32 adult Congolese migrants and explored beliefs, perceptions and lived experiences of migration, healthcare, vaccination and the COVID-19 pandemic. Reflexive thematic analysis generated two themes and a model conceptualising the vaccination decision-making process. Participants and community partners were financially compensated; ethics was granted by the University of London ethics committee (REC: 2021.0128). Participants highlighted the incompatibility of lockdown restrictions with their communal culture, which intensified feelings of exclusion and alienation. Concerns about COVID-19 vaccination were attributed to safety and effectiveness, partly informed by experiences and legacies of racial discrimination and exploitation. Inequality in the pandemic response and COVID-19 outcomes heightened participants' sense that their views and needs were being overlooked, and government sources and information were perceived as coercive. Our model depicts the interplay between institutional trust, belonging, and message perception, which shaped

**Data Availability Statement:** Data are available upon request due to ethical restrictions on sharing

data publicly. Please contact researchdata@sgul.ac.uk.

**Funding:** This work was supported by the National Institute for Health Research (NIHR Advanced Fellowship 300072) and seed funding from St George's Public Engagement Network. AFC is additionally funded by the Academy of Medical Sciences (SBF005\1111). SH acknowledges funding from the NIHR (NIHR300072; NIHR134801), the Academy of Medical Sciences (SBF005\1111), the Medical Research Council (MR/N013638/1), La Caixa Foundation (LCF/PR/SP21/52930003), UKRI/Research England, UK Health Security Agency (UKHSA), and the World Health Organization. AD is funded by the MRC (MR/N013638/1). The funders did not have any direct role in the writing or decision to submit this manuscript for publication. The views expressed are those of the author(s) and not necessarily those of the NHS, the NIHR, or the Department of Health and Social Care. The funder of the study had no role in study design, data collection, data analysis, data interpretation, or writing of the report.

**Competing interests:** The authors have declared that no competing interests exist.

participants' vaccination decisions and led to (non-)engagement with COVID-19 vaccination. This research enhances understanding of how social and contextual factors may influence migrants' engagement with health interventions. It underscores the importance of partnering with migrant communities to understand their needs in context and co-design tailored interventions and inclusive messaging strategies that promote trust and belonging. Implementing systemic changes to address structural inequalities will be crucial to create an environment that supports engagement with health-protective behaviours and enhances health outcomes among migrant communities.

## Introduction

Migrants suffered disproportionately during the COVID-19 pandemic, facing heightened health risks [1–4], poorer health outcomes [1,5–8] and lower vaccine uptake rates [9–15], compared to non-migrant communities. There is an urgent need to critically examine the structural factors underpinning these inequitable outcomes, to guide the design of inclusive interventions and policies which can advance health and vaccine equity. While trust has emerged as a key factor shaping vaccination decisions, especially among marginalised groups [16,17], the intricate dynamics of trust within specific migrant communities, and the underlying structural and social determinants, remain relatively unexplored [18,19] or oversimplified [19,20]. To bridge this knowledge gap, this participatory research study will explore the influence of context, lived experiences and notions of belonging on Congolese migrants' vaccination behaviour.

Trust can be understood as the confidence individuals have in their interactions with others, institutions, and societal systems [21], and is influenced by their beliefs, values and lived experiences. Several theoretical concepts can help us to understand how migrants conceptualise trust and feel connected to the world around them. Social identity theory [22,23] and belonging [24] provide two means of understanding the range of processes, including perceptions, attitudes and behaviours, that can contribute to identity. The latter concept offers a more nuanced framework and proposes that belonging (a sense of being 'at home' [25], feeling safe and connected) is formed through diverse attachments and memberships that, when denied, lead to exclusion and marginalisation. One of the ways this exclusion manifests is through othering, a process of exclusion rooted in power dynamics [26,27], and which is notably used as a political tool to evade responsibility for migrants and the globally displaced. In Australia, research with black African migrants and refugees living in Queensland highlighted the detrimental effects of othering practices and marginalisation on their sense of belonging to Australian society [28]. Similarly, the emphasis on race and ethnicity as COVID-19 risk factors in the UK led to increased stigmatisation and alienation of individuals of Black African, Caribbean and South Asian descent during the pandemic [29]. Crenshaw's intersectionality theory [30,31] provides a lens through which to understand the layered exclusion faced by migrants, by considering their overlapping social categorisations or identities (for example, based on migrant status, ethnicity, class and gender) and the interconnected systems of oppression, domination, or discrimination they create [31]. Applying these concepts to our research may facilitate a more nuanced understanding of the context in which migrants' health and vaccination decisions are made, providing us with the insight to develop more responsive and tailored interventions. Box 1 provides a preliminary overview of the context in which members of our study population migrated to, and live in, the UK.

## Box 1. History of the Democratic Republic of the Congo and Congolese migration to the UK

The Democratic Republic of Congo (DRC) has a complex history, with the Congolese people subject to exploitation, cultural repression, and systematic oppression under Belgian colonial rule from the late 19th to mid-20th Century. This era promoted harmful racial hierarchies and ideas of superiority of Europeans over Africans, while the economic exploitation of Congo's natural resources perpetuated a narrative of exploitation which continues to shape Congolese beliefs and values [32]. The country's struggle for independence from Belgian colonial rule is said to have fostered a sense of nationalism and pride among Congolese people [32]. However, this period also exacerbated ethnic tensions between the country's more than 200 different ethnic groups, potentially creating a complex picture of identity [32].

In recent decades, Congolese migration to the UK has been spurred by political instability, armed conflicts, economic challenges, and human rights abuses in the DRC. Consequently, a significant proportion of Congolese migrants in the UK arrived as refugees and asylum seekers, and they were the fourth most common nationality resettled in the UK between 2010 and 2021 (1774 people). Despite their complex history, relatively little is written about the Congolese diaspora in the UK. A 2006 report by the International Organization for Migration highlighted specific challenges, including language barriers, cultural adjustments, discrimination, difficulties accessing key services (housing, healthcare, employment), and challenging asylum processes [33]. These have led to prolonged periods of uncertainty, impacting their ability to integrate into UK society [33]. The reported reliance of these communities on the support of Congolese community organisations and non-governmental organisations [33] gives further indication of the detrimental effects of these challenges on their sense of belonging and inclusion within UK society, potentially shaping their willingness and ability to engage with the wider system.

Research in this area to date has been relatively siloed, focusing on migrants' constructions of belonging and identity, experiences of discrimination and marginalisation, or their health outcomes, but not causally linking these factors. Moreover, the overly simplistic explanation for behaviour based on "cultural differences" has been criticised for perpetuating and legitimising inequality relations [34]. However, recent studies have begun to consider the wider context shaping migrants' health decision-making. A study among Black African and Caribbean communities in the UK, for example, found that the uncertainty of the pandemic, combined with contemporary and historical mistrust and a lack of identity-aligned messaging, contributed to belief in conspiracy theories and low engagement with COVID-19 health-protective behaviours [18,29]. A study in Japan found that the social integration of migrants positively correlated with their COVID-19 vaccine acceptance, underscoring the importance of inclusion and community engagement [35]. Building on this recent work and the different understandings of these concepts, this study aims to explore the contextual factors shaping migrants' COVID-19 vaccination decision-making through an in-depth study with Congolese migrants in the UK. This approach moves beyond the limitations of the traditional 'information deficit model' [36], providing a comprehensive understanding of how migrants' unique contexts, worldviews and intersecting identities shape their attitudes towards and engagement with science [37], and paving the way for more tailored, culturally-embedded and inclusive interventions to address their unique needs.

## Methods

### Study design

A community-based participatory research (CBPR) study was conducted with Congolese migrants in London, UK in 2021–2022, led by a community-academic partnership ('the coalition', see below). Participatory research approaches emphasise doing research "with" rather than "on" people, employing collaborative methods that address power imbalances and value experiential knowledge [38]. While seen as promising for fostering more inclusive research involving migrant and marginalised groups, their implementation in this context remains limited [39]. This study originated from Congolese migrant community members identifying unmet needs and concerns around COVID-19 vaccination within their community, coupled with the academic partner's interest in exploring existing evidence gaps related to vaccination beliefs and behaviours among migrant populations [40–42]. The coalition's shared aim was to co-design tailored vaccination interventions with the Congolese community (and the published protocol and co-design study findings can be viewed elsewhere [43,44]). Nested within this work, qualitative, semi-structured in-depth interviews were conducted with Congolese migrants to explore their beliefs, perceptions and lived experiences of migration and healthcare in the UK, the COVID-19 pandemic, and routine and COVID-19 vaccination, which are reported here. The coalition co-designed and pilot-tested the topic guide and jointly decided on the data collection approaches and all aspects of the study design. The Standards for Reporting Qualitative Research (SRQR) were followed [45] (S1 Checklist).

### Setting and population

The study was set in Hackney, a diverse borough of London, UK, where over 89 languages are spoken and around 40% of the population come from Black and Minority Ethnic Groups, as defined by the UK Office for National Statistics [46]. Hackney was the 11[th] most deprived local authority in England based on the Indices of Deprivation 2015 [47]. It is thought to host one of the UK's larger populations of Congolese migrants [43,48]. A single nationality migrant group was involved for an in-depth, culturally situated understanding of the research topic, although it was recognised the sample differed across a variety of other criteria. The target population was Congolese adult migrants living in or around Hackney (specific inclusion/exclusion criteria are shown in Table 1).

### Community-academic coalition

The all-female coalition included an academic researcher with a background in community health research and lived experience of migration (AFC), three Congolese founders of a local community organisation in Hackney (LML, MLK, SN), with lived experience of migration and strong links with the local Congolese and Hackney communities, and an experienced

Table 1. Inclusion and exclusion criteria of study participants.

| Inclusion criteria | Exclusion criteria |
|---|---|
| • Born in the Democratic Republic of Congo (DRC).<br>• Aged 18 or above.<br>• Currently residing in the UK.<br>• Willing and able to give informed consent. | • Not migrant as per earlier definition.<br>• Not born in the DRC.<br>• Below the age of 18.<br>• Temporarily in the UK for holiday, visiting friends/relatives, or other reasons.<br>• Lacking capacity to consent, as determined by the mental capacity act framework. |

community outreach coordinator from Hackney's voluntary and community support sector agency (CH). The coalition met regularly to allocate roles and responsibilities and plan the study (described elsewhere [43]). As the three Congolese members of the coalition expressed an interest in conducting the interviews but had no prior research experience, the academic researcher (AFC) trained them in qualitative interview techniques and study design and the outreach coordinator (CH) provided training in facilitation skills two months prior to starting the study. The team had ample time to learn and practice their skills through role-play and pilot testing the topic guides.

## Participant selection and recruitment

The study aimed to recruit approximately 30 adult migrants who met the specified inclusion criteria (see Table 1). Participants were recruited (14/01/22-18/03/22) by the Congolese members of the coalition (LML, LMK, SN) using co-designed flyers, word-of-mouth, and snowball sampling techniques. All individuals approached had the opportunity to opt out of the study, and this did not preclude them from participating in later study activities, such as the dissemination event. They received a participant information sheet explaining the study and their rights, which was also explained to them verbally, at least one week in advance of the interview, and had the opportunity to ask questions before deciding whether to participate.

## Ethics and informed consent

The study was approved by the St George's University of London Research Ethics Committee (REC reference 2021.0128). All participants provided written informed consent prior to participating.

## Data collection

Semi-structured in-depth interviews were conducted in January-March 2022 and were done face-to-face in private meeting rooms by four members of the coalition (AFC, LML, LMK, SN). CH provided logistical and project management support. Interviews were conducted in Lingala, French or English (LML, LMK, SN are tri-lingual, and AFC used a professional telephone interpreter as required; most interviews were done in Lingala) and lasted 15–50 minutes. Interviews explored experiences, beliefs and knowledge of vaccination (including COVID-19, routine, selective and catch-up vaccinations), health-seeking behaviour, experiences of the NHS, preferred health information sources, and suggestions for improving health services. Topic guides were developed iteratively over the course of the study. Interviews were audio-recorded by Dictaphone and transcribed and translated verbatim by a professional translator. Transcripts were checked for accuracy (with community partners facilitating member checking and validation) and pseudonymised, and audio recordings were destroyed after transcription. Handwritten field notes were incorporated into final transcripts. Sociodemographic data were collected at the time of interviews from participants using a standardised form. Additional data and insights about the local, socio-cultural, and historical context, and Congolese culture, customs, and preferences were collected using optional poster walls which participants were invited to contribute to anonymously outside of interviews. Optional post-interview feedback forms were also collected. Participants received a £20 gift card for taking part and were reimbursed for travel costs and provided with childcare when required. Specific decisions were also made to encourage attendance and create a more welcoming and informal environment for participants, with the guidance of the Congolese coalition members. For example, interviews were held at a local community centre close to the local market, timed to

coincide with market days, and organised as part of 'community days' celebrating Congolese culture, during which information was shared about local support services.

### Data analysis and theoretical framework

A six-stage reflexive thematic analysis [49,50] was followed to explore patterns and subjective experiences across the dataset. A moderate constructivist theoretical framework [51] was used, recognising the roles of the researchers and participants in the co-construction of meaning [52]. Theoretical flexibility meant the analysis could be informed by critical and social justice approaches concerned with 'giving voice' to the participants and their lived experiences and locating these within wider political, historical, and sociocultural discourses. Transcripts were uploaded into NVivo12 software for qualitative analysis. The first author (AFC) completed stages 1–3, using a mixture of semantic and latent coding, and then discussed initial themes with the coalition (LML, LMK, SN, CH), TV and FK in a process of critical engagement to deepen the analysis, develop and refine themes (stages 4–5). During this stage, important patterns in the data relating to belongingness and identity were noted, which framed the onward analysis. Finally, a model was developed to theorise the links within and across the dataset and showcase how individuals' lived experiences may lead to engagement with specific health behaviours. AFC had regular conversations with co-authors (particularly TV and FK) throughout the analytic process and kept reflexivity notes. These were crucial to the development of the final themes and model and ensured subjectivity was acknowledged. Several iterations of theme names and definitions were reviewed to ensure their clarity, scope, and fidelity to the overall storyline. Consequently, the four initial candidate themes were refined into the two final themes presented below.

### Findings (analysis and interpretation)

Thirty-two participants were included in the study (descriptive characteristics are shown in Table 2), of whom most were female (24, 75%) and refugees or asylum seekers (19, 60%). The inclusion criteria were expanded during the study to include two (6%) participants who were born in Angola but identified as Congolese, recognising the geo-political limitations of the original criteria. Participants had a mean age of 52.6 years (standard deviation, SD, 11 years) and had lived in the UK for mean 14.3 years (SD 7.5 years). Most spoke Lingala (28, 88%) or French (20, 63%); few spoke English (10, 31%) and 15 (47%) considered themselves to have limited English proficiency (unable to read or write). All (100%) were registered with a GP.

The analysis generated two themes, which informed the model depicted in Fig 1. The model aims to theorise the engagement or non-engagement with health-protective behaviours, in this case COVID-19 vaccination, of Congolese migrants by considering the influence of context and lived experiences on their health decision-making.

### Theme 1: Belonging and identity in the wider context of lived experiences of migration

This theme first explores the intersecting social categories, systems and power structures shaping our participants' sense of belonging [24]. Next, it considers the impact of the COVID-19 pandemic on participants and explores how the policy response challenged notions of belonging and identity and elevated participants' alienation and mistrust towards Government and wider social systems.

Participants' narratives presented a layered picture of belonging. Communality, respect for elders, and being connected to their religious faith emerged as important aspects of their social identities. The significance of faith was noted and reflected in their use of biblical references to

Table 2. Characteristics of qualitative interview participants (n = 32).

| Characteristic | n (%) |
|---|---|
| **Migrant status** | |
| Seeking asylum | 6 (19%) |
| Refugee | 13 (41%) |
| British (naturalised) | 6 (19%) |
| Prefer not to say | 5 (16%) |
| Other visa | 2 (6%) |
| **Age in years, mean (SD)** | **52.6 (11.0)** |
| 25–49 | 13 (41%) |
| 50–64 | 15 (47%) |
| Over 65 | 4 (13%) |
| **Gender** | |
| Female | 24 (75%) |
| Male | 8 (25%) |
| **Time since arrival in the UK (years), mean (SD)\*** | **14.3 (7.5)** |
| 0–9 | 6 (19%) |
| 10+ | 22 (69%) |
| 20+ | 9 (28%) |
| Not available | 2 (6%) |
| **Country of birth** | |
| Democratic Republic of Congo or Republic of Congo^ | 30 (94%) |
| Angola† | 2 (6%) |
| **Religion** | |
| Christianity | 32 (100%) |
| **Marital status** | |
| Single | 18 (56%) |
| Married | 10 (31%) |
| Other | 4 (13%) |
| **Currently have children <16 years of age living in household** | |
| Yes | 15 (47%) |
| No | 17 (53%) |
| **Languages spoken** | |
| Lingala | 28 (88%) |
| French | 20 (63%) |
| English | 10 (31%) |
| Other (Kikongo, Portuguese) | 3 (9%) |
| **Limited English proficiency (self-reported, cannot read or write in English)** | |
| Yes | 15 (47%) |
| No | 14 (44%) |
| No response | 3 (9%) |
| **Registered with GP** | |
| Yes | 32 (100%) |

articulate their beliefs. Participants' cultural identities were strengthened through attachments to Congolese culture and ways of life, including social gatherings, gospel music, creative expression, and local community ties. These attachments fostered feelings of safety and inclusion within their in-group, offering a refuge of shared experiences and values. However, participants also shared their perceptions of not being embraced by, or belonging to, British society. These perceptions were informed by experiences of exclusion and othering, rooted in language

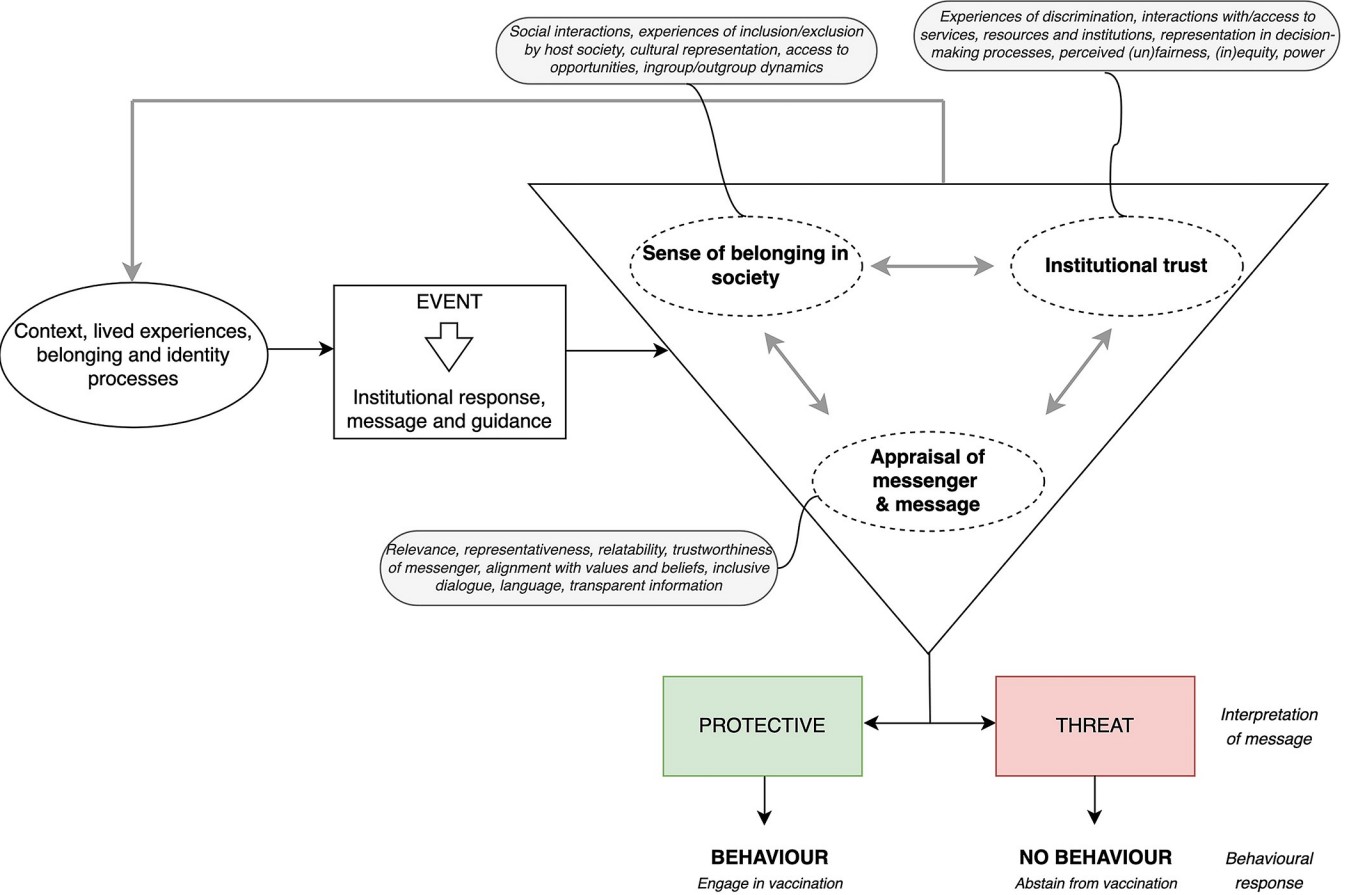

**Fig 1. Model to understand engagement or non-engagement with COVID-19 health protective behaviours/vaccination in Congolese migrants in the UK.** The "EVENT" represents the COVID-19 pandemic. Institutional trust reflects confidence in authorities, sense of belonging represents connectedness to wider society and in-groups, and perception of messenger/message relates to trustworthiness and relevance of information based on their worldview.

barriers, citizenship status (or labelling/othering based on legal status), discriminatory practices, and the challenges of navigating the system as a migrant. This was particularly evident when it came to navigating the healthcare system, where they highlighted examples of receiving sub-standard care and having their eligibility questioned. Some participants said they had sought alternative treatment or withdrew from seeking further care because of this, while others expressed a sense of injustice, particularly given their contributions through taxation. These shared experiences may have fostered a sense of comradery and internal belonging (with their in-group or 'insiders') while deepening their sense of alienation from wider society (their perceived out-group or 'outsiders').

*"In my case, because I work a lot with the community, yes, some people are asylum seekers and there's a discrimination when they turn up to a GP to register. They ask them for so many documents, your passport, proof of address and when they don't have that it's just putting them off." P5*

\*\*\*

*"If we go again [to] the GP in this country, when you try to phone them, maybe have to take you two hours, three hours, before to pick up the phone. And then when they pick up the*

*phone in reception, okay, they give you now appointment. Appointment may be two weeks or three weeks before to see you. I don't know. [. . .] Is embarrassing. For now, the GP, I think, is a joker. [. . .] When they come to see you after that, they give you a paracetamol. What's going on? We work in this country. We pay the tax. . . Why the thing must go like that? Why?" P11*

Participants described varying responses to their experiences of exclusion. Some responded through self-protective mechanisms such as withdrawal, while others adapted and formed new attachments and memberships which secured their inclusion, and potentially increased their sense of belonging in society or connectedness to wider society (the out-group) (Fig 1). Furthermore, beliefs and perceptions regarding belonging and identity were dynamic and appeared to evolve over time and across generations. One participant shared that when they first arrived in the UK, they had been told, "this country is English country, to understand and to be free you need to learn [English]" (P26). It is unclear who their interaction was with, but their further comments suggested they now conformed to this mindset:

*"Not Congo, I'm not living in Congo at the moment, now I'm in here in England. I need to follow the instruction from here. I need to follow the news from here to get better life. So that better life we need to share with people who doesn't understand something like this. They don't like to understand because many African they not respect the law. Yes, many African they don't want to respect the law. I don't know why." P26*

Reflecting on the pandemic, participants highlighted the incompatibility of restrictions with their communal culture and explained that lockdowns and social distancing measures had impacted their ability to be together, which had led to sadness, loneliness, and uncertainty, as well as the perception of 'being watched'. The loss of loved ones to COVID-19 further compounded their distress. Drawing on the framework of social identity theory [22,53], the enforced separation during the pandemic may have disrupted participants' collective and self-identities and sense of belonging at a more profound level by compounding their experiences of marginalisation.

*"Yes, it's a big problem because now we cannot meet each other and then most people, they lost their relatives. And then it divided people, that's what I can say because we Africans need to be together, but because of Covid we cannot be together. No party, no group, no church. We like to go to the church to pray, to meet with people but we don't do it anymore. [. . .] They were not allowed to come because the government is watching them." P21*

\*\*\*

*"No, you know, Congolese people we are different, I am telling the truth. Some people care, and other don't. If you are telling them to respect social distancing, they will tell you that we are community people. We are different." P18*

During this period of uncertainty, participants drew on familiar belief systems, such as their Catholic faith. Several participants collectively identified with the need for 'proof' when faced with uncertainty; the absence of which may have heightened their sense of vulnerability.

*P18: "Congolese people are like Thomas who wanted to put his finger in Jesus wound before he would believe that Jesus has been resurrected.*

I: You mean Congolese people need proof?

*P18*: *Yes, they need proof. . ."*

Some participants highlighted the impact of the pandemic on their spiritual journey and relationship with God, referring to the Ignatian concept of discernment of spirits–judging good from evil–guiding their worldview and decision-making process. Navigating the crisis of the pandemic without regular access to spiritual guidance may have intensified participants' perceived need for spiritual discernment. This is reflected in how they conceptualised the pandemic and their (or others') engagement with government messages and guidance. Some placed their faith above governmental matters, while others saw non-compliance with health guidance as a form of defiance against divine orders.

*I: In your view, how has the government managed the Covid-19 pandemic?*

P20: I don't have an opinion about the government because I am more engage(d) in serving God than the government business.

***

I: Why do you think those people they don't want to listen to the government or the advice?

*P26*: *Is [because] they're bad spirited [. . .] It's like God give us intelligence and such, what does God give to the nurse, to the doctor, is like a good spirit to help us [. . .] So if you don't listen to the people who want to save your life, it's like you are abusing the order from the heaven. Sometimes, it's like when we are praying, we pray for those people.*

## Theme 2: Evaluating and interpreting health information

This theme considers how participants interpreted and evaluated health information: (a) interpreting the message, and (b) evaluating the messenger: representativeness and trust.

**A) Interpreting the message.**   Nearly all participants perceived vaccines in general as protective ("Vaccines are good to protect against diseases") and routine ("We take vaccines from birth") but expressed mistrust of COVID-19 vaccines specifically. They largely attributed this mistrust to rumours, including misinformation, and muddled messaging about COVID-19 that were spread through social media, social networks (friends and family), and government messaging.

*"Yes, people know that vaccines protect, but based on the information they are reading on media on TV, it's putting people off." P5*

Many participants experienced dilemmas while evaluating health information and deciding on a course of action, due to the presence of widespread rumours, misinformation, fear, and suspicion about the COVID-19 vaccine. They expressed uncertainty about whether to get vaccinated, driven by a perception that the vaccine may pose risks to black people. These concerns were further compounded by worries about fairness and the potential for discrimination. A crucial factor influencing their decision-making process appeared to be their personal biography as refugees, having come to the country for protection. This unique background added an additional layer of complexity to their risk assessment and decision-making. Many participants acknowledged the prevailing confusion and misconceptions within African and Congolese communities regarding the safety and effectiveness of the vaccine.

*"It was not easy for me [to get the vaccine] because there was so many rumours and I was questioned myself if do I have to take it or not. We came in this country to seek protection. I*

*just realised that white people are receiving their vaccinations, its means that not only they want to kill only black people. All of us are standing in the same queue to receive the same vaccine. It means that all of us are going to die? There has been lot confusion about negative rumours concerning covid-19 vaccine. African in general and especially Congolese people has wrong judgement about the vaccine. We are not doctors but we [are] searching about vaccines. It was not easy to take vaccines because of negative rumours and publicity saying that if you receive the vaccine you are not going to live for the next 3 months. Another person placed his phone on his arm to show how the phone was stuck after receiving his jab. How come a mobile phone can stick on somebody's arm?"–P4*

Participants shared concerns and rumours [40,54–56] reported by other migrant and ethnically minoritised groups about the origins and consequences of COVID-19 vaccines/vaccination, including that they were designed by people or systems (scientists, the NHS, the government, Bill Gates) to cause harm, kill, reduce, or spy on the global population, and black and African populations specifically; or that they had demonic or apocalyptic associations. They also expressed concerns about safety and side effects, which originated from the aforementioned conspiracy theories, public information (e.g. clotting risks associated with the AstraZeneca COVID-19 vaccine), knowledge of their own risk factors, and personal experience of side effects. Some participants believed that the evidence of adverse events and the frequent changes in government guidance and recommendations, such as the need for additional doses and boosters, indicated that the vaccine was not safe or ready when it was first rolled out. They contrasted this with the flu vaccine, which they considered to be safer:

*"People are not really reluctant to have flu vaccines because they are convinced that it has been experimented in people for many years and it's safe for them to take it, but this one [COVID-19 vaccine], there are a lot of things that have been said."–P5*

Changes in guidance and perceived lack of transparency by the government were off-putting to many participants and appeared to erode trust in the messenger. The lack of clarity and certainty in messages resulted in confusion and led some participants to question if something more sinister was being hidden behind the vaccine. Some also considered the vaccine's inability to prevent infection as a scientific failure. Participants questioned the clarity, relevance, and representativeness of messages and gave examples of ways the messages didn't reflect their reality or circumstances; for example, two participants dependent on shift work said they had wished they had known more about the vaccine's side effects before getting vaccinated, so they could better plan their schedules.

*"I was not sure about the research carried of the vaccine as researchers themselves had double-speak [and] were not sure about their own work. I was scared and reluctant about the vaccines because I was confused with the information from research. The vaccine prevents you not to catch illness but it doesn't means that you may not be infected. There was a lot of confusion, and this was the reason I was not ready to be vaccinated. I was not sure because scientists were not clear in their language." -P6*

Participants indicated that in the absence of clear information or proof about the consequences of vaccination, they needed to seek out answers independently, as part of their risk evaluation and decision-making process. This centred on whether a course of action (e.g. vaccination) could be deemed 'protective' or a 'threat'. Events which heightened the perception of vaccination as a threat were more prevalent and included vaccine scares reported by the media

and relatively rare instances of vaccine-derived poliomyelitis (notably, DRC is one of few countries with recurring vaccine-derived polio outbreaks), which participants recalled from polio vaccination campaigns during their childhood. For example:

> *"Some children have become disabled after receiving polio vaccine. [. . .] [They are afraid] because the side effects of vaccine have caused to their children to become disabled, and they don't want again to take the risk."–P2*

Many participants said they had developed hypertrophic or keloid scars from vaccination in childhood, which had made them associate vaccination with pain and fear. They also alluded to historical racist and unethical medical practices against African people and expressed concerns that they were still being used as 'guinea pigs' into the present day, referencing racist remarks by French doctors about testing coronavirus vaccines in Africa [57].

The principles of heuristics (mental shortcuts to making quick decisions) can help to explain decision-making processes under uncertainty [58]. Facing an overwhelming amount of contradictory and changing information, our participants may have experienced indecision and defaulted to believing in or following information that confirmed their worldview (confirmation bias), or that which was most readily available to them (such as misinformation and rumours). This may help to explain why individuals experiencing alienation and social exclusion may have been less likely to engage with COVID-19 vaccination (Fig 1).

**B) Evaluating the messenger: representativeness and trust.** *The source of the messenger* delivering the message was critical in determining whether it was trusted and followed by participants. Messages delivered by 'outsiders', including the Government, were generally perceived as misleading, unreliable, and coercive. Messages delivered by 'insiders', such as members of participants' own communities, were considered more representative, relatable, relevant, and trustworthy. Participants were more trusting of local community members who were known to them and represented their communities rather than somebody who might not know their community's histories.

Participants generally expressed mistrust towards the Government and felt they were being "pressured", "forced", and "pushed" by the Government to receive COVID-19 vaccines. Vaccine reminders were perceived on a negative spectrum, ranging from an annoyance to an indication of something more sinister that should be resisted ("It means that there is unknown information behind this." P4). One participant remarked, "I would do it voluntarily, but not by force." (P16), adding that the constant reminders had deterred them and heightened their suspicions of what might be hidden behind the vaccine. Comparisons were again drawn with the flu vaccine, which was felt to be uncontentious:

> *"People have never been forced to receive flu vaccine. If you don't want to take your flu jab, the GP will not force you to take the jab, but this (referring to the covid vaccine) has become an obligation." P4*

Although we did not explore vaccine mandates with all participants, those we spoke to felt they imposed excessive control over people's lives. One participant (P24) said they would "definitely" leave their job if their employer introduced mandatory COVID-19 vaccination. It was also apparent that some participants felt their views and concerns were being ignored and overlooked, and mandates were a way for the Government to avoid addressing or considering wider and more peripheral viewpoints and being used as a form of oppression and control.

*"The people in the power, the government, must listen to the voice of all the people antivax. After three months, you must take the vaccine [booster]. It's no good like that. And then COVID pass is no good. Before you get to the restaurant, you go to the events like that, you must to show something like that. It's not work like that. The people now limits. Life, now, is finished." P11*

One participant explained why mandates and reminders were so triggering and proposed ways of communicating that might be more effective and less likely to elicit an emotional response or feeling of being coerced among members of their community. Community members wanted reassurance, and for their concerns and reservations to be addressed, rather than perceived as unjustified.

*"Basically, to force someone is like he thinks you give him poison. When you force someone… like me, I like vegetables, but she doesn't like, he doesn't like vegetables. Why can you force him? You need to help him to understand that vegetable is not poison. […] So, when you force someone, it will get [him] thinking more things [like]: 'they kill me', 'they give me poison', 'they want to…' He will think [that] because no he doesn't like [it]. But if you help, you advise, you make him understand, like [with a] baby: 'don't do that', 'don't touch', 'this is a fire', 'don't eat this, it's a poison', until you will see him calm down [and] say 'Really, this thing is good, let me go and get it'."–P26*

For many participants, the Government represented the orchestrator of their reduced freedoms and the supplier of said 'poison'. As alluded to above though, some participants described having positive interpersonal relationships with their local social network, including GPs, managers, and neighbours, which were influential on changing their views and behaviour. For example, a few participants who had been initially hesitant about COVID-19 vaccination changed their mind after having a conversation with a trusted source or seeing a friend get vaccinated:

*"Yes, I spoke with my GP when the news came out that people were having blood clots and I was so scared, I cancelled my appointment two or three times. […] My GP advised me and said don't worry. […] That's when I was convinced to go and get the second one because I seek medical advice." P5*

\*\*\*

*"After I have seen too, some people around me, they got the vaccination and they advise me. I say, okay, I need to go get [it]. […] My wife too, got the vaccination. I send her to go get the vaccination after I got my one." P29*

Certain members of the community also took on the role of encouraging others to get vaccinated. This included individuals who had initially been ambivalent or cautious but had since had a positive vaccination experience. Unlike the official messaging deployed, these unofficial community role models used approaches that were perceived as being more representative, relevant, and resonated with the community and their values. One participant described how they encouraged their community "not by force", and appealed to values of community, care, protection, and safety:

*"We keep advising them not by force, but patiently to tell them respectfully, to explain to them it's like this, it's important, it's for saving life, saving our kids, saving everything in our community. Something like this." P26*

Others directly addressed the specific concerns and causes of mistrust that people raised, and which they could relate to, such as hidden Government motives or malintent stemming from histories of repression, and sought to debunk them:

*"Some for them, it's a scare. But I advised them, I said to them, it's nothing. Because look at it. People is thinking wrong. They say maybe it's a poison. I have advised them. I said, no. Look at me. In England, it's not like Africa. Because the government for England is work for their people. Is work for their community. It can't kill anyone, everyone in the same [boat] about the vaccine." P29*

### Linkage to model

Fig 1 provides an explanation for how participants' context, lived experiences and sense of belonging shaped their interpretation of information and their behavioural response to COVID-19 vaccination. The triangle depicts the interplay between institutional trust, belonging, and the appraisal of messages and the messenger. Our data suggest these factors vary between individuals and can change over time (shown by the grey arrow). When these factors align positively, individuals adopt health-protective behaviours (perceiving them to be protective), like vaccination. Conversely, if individuals feel alienated, mistrust institutions, and messages and messengers do not reflect their values or needs, they may reject the information (perceiving it to be a threat) and not adopt the desired health-protective behaviour.

## Discussion

This study explored how the lived experiences of Congolese migrants in the UK shaped their sense of belonging, trust in institutions, message perception, and ultimately their vaccination decision-making during the COVID-19 pandemic. Participants described how experiences of discrimination, exclusion from healthcare, and pandemic-related restrictions heightened their feelings of alienation from British society. Awareness of medical exploitation of Black Africans and vaccine-related fears and scares contributed to concerns about the safety and effectiveness of the COVID-19 vaccine. Additionally, perceptions of government corruption in the DRC, and a lack of clear, relatable or representative messages from UK officials, reinforced mistrust of authorities. The conceptual model depicts the interplay of these factors in shaping participants' vaccination decisions, leading to (non-)engagement with COVID-19 vaccination, while also providing a framework for understanding broader health-related decision-making. These findings emphasise the profound impact of personal experiences and worldviews on vaccination decisions, particularly among intersectionally marginalised populations, and underscore the importance of addressing structural inequalities to strengthen vaccination uptake.

Our findings build on recent research, such as Vandrevala et al. (2022), which highlighted how the crisis of the pandemic, combined with historical and contemporary mistrust, provided a context for alternative conspiracy narratives to thrive in UK black communities [18]. Our study findings lend support to this interpretation, while further underscoring the pivotal role of context and lived experience in shaping migrants' belonging and identity processes and levels of institutional trust, and subsequently, how health messages are received and acted upon. Bury's notion of biographical disruption [59] may also help us to understand the impact of the pandemic on our participants' sense of belonging and their decision-making. Biographical disruption aims to describe the influence of a significant, sudden event on the course of a person's life, which Bury argued occurs in three ways: 1) disruption of 'taken for granted' assumptions and behaviours, creating heightened awareness of our bodily state, 2) disruption of explanatory

frameworks, leading us to re-think our biography and question our sense of self and future trajectory, and 3) disruption of the way we deploy our resources, physically (time and effort) and socially (activities we pursue; financially) [59,60]. We found that the pandemic heightened participants' awareness of their actions, prioritising behaviours vital for their safety or security. It prompted them to ask questions about their biography, such as why this was happening to them, and what had caused it; and it changed the way they acted and behaved. Applying this lens may deepen our understanding of our participants' responses to the pandemic and their COVID-19 vaccination choices and support the design of more personalised vaccination and health interventions. Moreover, these insights may help define responses to help individuals cope with enduring life changes beyond the pandemic.

Several qualitative studies exploring beliefs about COVID-19 vaccination and other vaccinations have identified concerns about safety and effectiveness and fear of side effects [40,54–56,61]. These were also reported in our study; however, we also noted novel factors specific to our population which might help to explain how these concerns originated. Many participants in our study associated vaccines with pain partly because of the keloid scarring they had developed after receiving childhood immunisations, particularly BCG. Extensive literature suggest that keloids form more readily in dark compared to light skin, cause pain and distress, and can form at vaccination sites [62]. This may help explain why fear of pain and side effects were frequently mentioned as a barrier to COVID-19 vaccination in this population, and perhaps also why participants expressed wanting to know in advance more about what to expect from getting vaccinated. Several participants experienced mass immunisation campaigns as children in the DRC, where the emphasis may have been on 'jabs in arms' as opposed to informing recipients or providing aftercare. Our findings highlight how the medical experiences of black and racialised people are overlooked in the medical literature and in informing medical practice. Greater consideration must be given to how specific medical practices can shape collective beliefs about vaccines and the medical institution over generations, especially among groups who already share beliefs that they have been historically exploited by medical authorities, with clear communication of risks and implementation of alternative measures where possible. Our participants also had relatively specific knowledge and views about vaccination, drawn from personal and group experiences. For example, participants highlighted cases of vaccine-derived polio, including paralysis, and international examples of vaccination scares. Being a migrant population with close links to a country with history of mass vaccination campaigns and circulating vaccine-derived poliovirus strains, these references should perhaps not be unexpected, and may help contextualise participants' fears. These examples highlight the importance of understanding population demographics and migration patterns and considering the influence of past experiences, histories, and context on values and behaviour when designing and implementing public health interventions, particularly for migrant populations. They also underscore the role of governments in promoting public trust and confidence, and of demonstrating their competence through proactive, transparent, and comprehensive communication with populations [63].

Participants' mistrust of Government and authority appear to stem from historical and more recent experiences of injustice and oppression. Depending on the timing and context of their migration (noting that around 60% of participants were asylum seekers and refugees), experiences of violence and war in Congo and marginalisation and discrimination as newcomers in UK society will have likely shaped their worldview. Their perception of vaccine reminders and recommendations as coercive is an important finding and suggests vaccine mandates or other enforcement-based public health measures (which are a recommended strategy for achieving high vaccine uptake [64]) would not be suitable to implement in this population (nor populations with similar experiences, beliefs, and value systems), as they would likely

further undermine trust. It also highlights the importance of considering the cultural relevance of interventions during their design (prior to roll-out) and involving members of represented groups in tailoring their design to ensure they are appropriate and acceptable for the intended audience(s). These findings align with other recent work, for example, a US study of the effect of COVID-19 vaccine mandates on vaccine attitudes and behaviours concluded that mandates are unlikely to change vaccination behaviour overall and may deter considerable percentages of people from engaging in activities where vaccines are mandated [65]. A second US study reported that more Black respondents found hypothetical vaccine mandates for adults unacceptable compared to non-Black respondents [66]. Our findings also corroborate and build on the hypothesis of Tankwanchi and colleagues (2021) who suggested, in their rapid review of literature reporting on vaccine hesitancy in migrant communities, that experiences of xenophobia, marginalisation, and discrimination in host countries diminish migrants' trust in the health system and may exacerbate vaccine hesitancy along a pathway of social exclusion [67].

Recent literature exploring reasons for vaccine hesitancy among diverse groups has proposed that aligning pro-vaccination messages with the moral values and intuitions that people endorse may be more effective than vaccination mandates. Of note, a UK study [68] found that individuals who more strongly endorsed the moral foundation of liberty, which prizes freedom, choice, and individual rights [69], tended to be more vaccine hesitant. Members of the Collaboration on Social Science and Immunisation (Australia) also recently proposed that less coercive, trust-promoting measures should be prioritised over mandates, along with efforts to understand and address context-specific factors [70]. Future research could involve exploring how migrants' unique contexts give rise to the endorsement of specific moral values. These insights could shed further light on the processes underlying vaccination behaviour, aiding the development of vaccination messages that integrate both context and value systems. This could be relevant to designing strategies to improve uptake of catch-up vaccinations in adolescent and adult migrants, which is an increasingly important policy area requiring further planning [71], as well as other health promotion strategies, and in future public health crises.

Overall, the findings of this study provide new insights into the importance of inequalities in understanding responses to public health crises, reinforcing the argument for increased research on this topic and for integrating the reduction of systematic inequalities as a fundamental component within disaster planning [72]. Our study has revealed that public health interventions can impact beyond health and interact with and influence notions of belonging in communities. A key recommendation is for local health and care partners, such as those within integrated care systems (ICSs) in England, to work closely with communities to understand their health and vaccination needs in context, involving them in decision-making, and jointly designing, implementing and evaluating services and interventions. Additionally, local populations should be involved in developing contextualised risk communication and community engagement plans to bolster resilience and preparedness in future emergencies. In parallel, governments must proactively address structural inequalities, build trust, and promote social cohesion and justice. In the UK, central government departments and bodies such as the UK Health Security Agency (UKHSA), Department of Health and Social Care (DHSC), Office for Health Improvement and Disparities (OHID) and NHS England should ensure sustained allocation of resources to regional teams to enable the commissioning of locally tailored and trusted programmes and services. Across government, there should be a firm commitment to embrace anti-racist approaches and whole systems thinking, supporting measures that redistribute power and resources and address upstream factors. By prioritising these actions, governments may be able to foster stronger connections with their diverse populations and address disparities in health outcomes more effectively.

### Strengths and limitations

Aligning with other initiatives aiming to address health inequalities [39,73–76], a strength of this study was its participatory, power-sharing approach, which provided members of Congolese community-based organisation with training and funding to lead a study exploring a public health issue of importance to their community. Having trusted members of the target population lead and conduct the study may have also contributed to the openness of participants and the rich data generated. Our theoretical and conceptual approach allowed us to explore our objectives through the language of our participants and locate their lived experiences within wider political, historical, and sociocultural discourses. Subsequently, our model may help improve understanding of migrants' health decision-making processes and lead to the development of tailored public health interventions which better consider their personal histories, cultural identities and lived experiences.

A limitation is the study's focus on a specific Congolese community in London, meaning that the specific historical and contemporary influences that informed their beliefs and behaviours may not be generalisable to other marginalised communities in different contexts. Our study also did not disaggregate by ethnicity; therefore, our participants may represent diverse Congolese ethnic groups whose unique histories are not adequately reflected here. Nonetheless, our overarching finding that health perceptions and behaviours are rooted in long-standing societal issues will be relevant for many intersectionally marginalised migrant populations, and has immediate implications for practitioners, policymakers and academics working to address health inequalities and design tailored public health interventions. Being an exploratory study, this research did not set out to explore notions of belonging and identity; rather, these themes were generated through immersion in the data and the inductively developed analysis. Future research may now build on and test our findings and theoretical approach more purposely. We discuss other strengths and limitations of our work in our accompanying intervention co-design paper [44].

### Conclusions

This study highlights the need to work closely with migrant communities to co-design culturally appropriate interventions that build trust and promote inclusion. Fostering this sense of ownership and trust may secure improved health outcomes among migrant populations. Immunisation and inclusion health teams within key government departments and bodies such as UKHSA, DHSC, OHID and NHS England, along with ICSs, must at the same time recognise the need to address structural inequalities and upstream factors to strengthen public health and immunisation programmes and reduce population health disparities.

### Supporting information

**S1 Checklist.**
(DOCX)

### Author Contributions

**Conceptualization:** Alison F. Crawshaw, Sally Hargreaves.

**Data curation:** Alison F. Crawshaw.

**Formal analysis:** Alison F. Crawshaw, Tushna Vandrevala, Felicity Knights, Laura Muzinga Lutumba, Sarah Nkembi, Lusau Mimi Kitoko, Caroline Hickey.

**Funding acquisition:** Alison F. Crawshaw, Sally Hargreaves.

**Investigation:** Alison F. Crawshaw, Laura Muzinga Lutumba, Sarah Nkembi, Lusau Mimi Kitoko, Caroline Hickey.

**Methodology:** Alison F. Crawshaw.

**Project administration:** Alison F. Crawshaw, Laura Muzinga Lutumba, Sarah Nkembi, Lusau Mimi Kitoko, Caroline Hickey.

**Resources:** Alison F. Crawshaw.

**Software:** Alison F. Crawshaw.

**Supervision:** Alison F. Crawshaw, Alice S. Forster, Sally Hargreaves.

**Validation:** Alison F. Crawshaw, Laura Muzinga Lutumba, Sarah Nkembi, Caroline Hickey.

**Visualization:** Alison F. Crawshaw, Tushna Vandrevala, Lusau Mimi Kitoko.

**Writing – original draft:** Alison F. Crawshaw.

**Writing – review & editing:** Alison F. Crawshaw, Tushna Vandrevala, Felicity Knights, Anna Deal, Laura Muzinga Lutumba, Sarah Nkembi, Lusau Mimi Kitoko, Caroline Hickey, Alice S. Forster, Sally Hargreaves.

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
