## [Decision Letter · Decision Letter 0]

21 Feb 2024

PGPH-D-23-02117

Navigating vaccination choices: The intersecting dynamics of institutional trust, belonging and message perception among Congolese migrants in the UK (A reflexive thematic analysis)

Dear Dr. Crawshaw,

Thank you for submitting your manuscript to PLOS Global Public Health. After careful consideration, we feel that it has merit but does not fully meet PLOS Global Public Health’s publication criteria as it currently stands. Therefore, we invite you to submit a revised version of the manuscript that addresses the points raised during the review process.

While both reviewers have evaluated your manuscript very positively, they have recommended a few editorial changes (see below).  Upon receipt of a revised version that considers/integrates the reviewers' comments, we will be able to move swiftly toward acceptance and publication of your manuscript.

We look forward to receiving your revised manuscript.

Kind regards,

Nora Gottlieb

Academic Editor

Journal Requirements:

1. Please provide additional information regarding the considerations  made for the migrants included in this study. For instance, please discuss whether participants were able to opt out of the study and whether individuals who did not participate receive the same treatment offered to participants.

2. Please update your online Competing Interests statement. If you have no competing interests to declare, please state: “The authors have declared that no competing interests exist.”

3. In the online submission form, you indicated that "Data are available on reasonable request from the study authors.". 

a) In a public repository, 

b) Within the manuscript itself, or 

c) Uploaded as supplementary information.

4. Please provide separate figure files in .tif or .eps format only and remove any figures embedded in your manuscript file. Please also ensure that all files are under our size limit of 10MB. You may leave the figure captions or legends in the manuscript.

Additional Editor Comments (if provided):

Reviewers' comments:

Reviewer's Responses to Questions

**Comments to the Author**

1. Does this manuscript meet PLOS Global Public Health’s publication criteria? Is the manuscript technically sound, and do the data support the conclusions? The manuscript must describe methodologically and ethically rigorous research with conclusions that are appropriately drawn based on the data presented.

Reviewer #1: Yes

Reviewer #2: Yes

2. Has the statistical analysis been performed appropriately and rigorously?

Reviewer #1: N/A

Reviewer #2: N/A

3. Have the authors made all data underlying the findings in their manuscript fully available (please refer to the Data Availability Statement at the start of the manuscript PDF file)?

Reviewer #1: Yes

Reviewer #2: No

4. Is the manuscript presented in an intelligible fashion and written in standard English?

Reviewer #1: Yes

Reviewer #2: Yes

5. Review Comments to the Author

Reviewer #1: Thank you for the opportunity to review this manuscript, which reports on rich qualitative data from a participatory study with Congolese migrants in the UK, and generates a novel theoretical model for engagement with health protective behaviours in this group.

The authors could consider including a short section in the introduction on participatory methods, to provide context for those less familiar with their approach. In particular, it would be useful to explain why this approach is most appropriate when working with marginalised communities such as migrants.

It may also be useful to provide a little more detail on work leading up to this study, in particular the CBPR project that it is nested within, and the context in terms of co-designing an intervention with the community. This will also help clarify the specific aim of this study compared with other wider goals of the CBPR project. I would suggest including this at the end of the Introduction, so that the study-specific aim leads directly into the Methods.

I also wonder if the paper could be strengthened by the addition of a Box/Figure/Table summarising the key recommendations that can be made as a result of this study. These are included and explained clearly in the Discussion, but drawing them out would help guide the reader to the direct policy implications of the research.

Some minor alterations:

• Lines 189-193: add initials for each researcher.

• Table 2: Add ‘No’ as well as ‘Yes’ under ‘Currently have children <16 years of age living in household’, for consistency with the rest of the table.

• Line 641 to 677: very long paragraph, consider splitting in two.

• References: ref 4 needs correction, lines 887 and 888 are blank (may be an error with ref 51).

Reviewer #2: This study is an important contribution to the field, providing a clear but nuanced description of vaccination decision-making among Congolese migrants in London. The first half of the Discussion is particularly strong, and the authors have done a great job at contextualising their findings. I have outlined some suggestions below which will hopefully improve the clarity in some sections of the paper.

TITLE & THROUGHOUT

I suggest considering whether the title and other aspects of the paper (e.g. the Intro section of the Abstract, the study aims) should be revised to ‘London’ rather than the UK. The authors mention in the Discussion that while Hackney is home to a large proportion of UK’s Congolese migrants, the study did not explore perceptions of Congolese migrants outside of Hackney/London, and therefore labelling the study with ‘UK’ suggests that the scope of the study and its aims were UK-wide.

ABSTRACT

Line 40-42: “but has been seldom explored”, unclear whether you’re referring to the factors or the interventions, consider re-wording e.g. “but these factors have been seldom explored”. Similarly line 67-69, run-on sentence that shifts away from present progressive tense mid-sentence – consider simplifying.

INTRODUCTION

Line 124-130: Outlining the aims of this study and then jumping to the ‘History of the DRC’ section before moving on to the Methods section is a bit odd. Perhaps flow could be improved if the ‘History of the DRC’ section could be included in a Box and placed somewhere earlier on in the Introduction section (and definitely before lines 124-130) so that the aims flows directly on to the methods.

METHODS

Line 172: Could the SRQR checklist please be included as a supplemental appendix?

Were the transcriptions and/or findings presented to the participants for member checking and validation?

FINDINGS

Figure 1: It would be quicker for the reader to connect the relevant factors box with the content of the triangle if the model layout was improved, e.g. position the two closer together and use ‘magnifying glass’ to connect the triangle to the relevant factors box

DISCUSSION

Lines 641-677: This paragraph is extremely long and the message of the paragraph gets lost. Consider splitting it in half at around Line 661-662 so that the first part focuses on interpretation of findings relating to institutional trust and the second half focuses on interventions/strategies/future research. Line 674 ‘Overall, the findings…’ also seems to fit better in the next paragraph which focuses on placing the responsibility on policy makers in improving future PH response.

Line 679-686: The ‘agents of change’ in this paragraph could be clarified e.g. rather than continuing to use terms like policymakers and practitioners, you could be specific e.g. is it the primary care networks’ and integrated care boards’ responsibility to evaluate/understand the needs of the community, or is it the responsibility of GP practices and other vaccine service providers? Similarly, who in government should ‘shoulder the responsibility’ of trust-building? From my experience, statements like this are often overlooked when they are presented to government agencies because it’s such a high-level and non-specific ask. For example in the UK context, should it start with key vaccine-related departments within peak bodies like UKHSA, DHSC, OHID and NHS England deploying resources to regional teams to then commission programmes of work from local (and more trusted) organisations? Of course the discussion cannot be very specific in this regard, but asking yourselves these questions like this may help you to think through your recommendations in more detail and target the message in this paragraph.

CONCLUSION

Following on from my comment above, consider how the final sentences can be a bit more targeted rather than saying ‘governments’ must do X, Y, Z.

6. PLOS authors have the option to publish the peer review history of their article (what does this mean?). If published, this will include your full peer review and any attached files.

**Do you want your identity to be public for this peer review?** For information about this choice, including consent withdrawal, please see our Privacy Policy.

Reviewer #1: No

Reviewer #2: No

---

## [Decision Letter · Decision Letter 1]

4 Jun 2024

Navigating vaccination choices: The intersecting dynamics of institutional trust, belonging and message perception among Congolese migrants in London, UK (A reflexive thematic analysis)

PGPH-D-23-02117R1

Dear Dr Hargreaves,

We are pleased to inform you that your manuscript 'Navigating vaccination choices: The intersecting dynamics of institutional trust, belonging and message perception among Congolese migrants in London, UK (A reflexive thematic analysis)' has been provisionally accepted for publication in PLOS Global Public Health.

Best regards,

Nora Gottlieb

Academic Editor

Reviewer Comments (if any, and for reference):

Reviewer's Responses to Questions

**Comments to the Author**

1. If the authors have adequately addressed your comments raised in a previous round of review and you feel that this manuscript is now acceptable for publication, you may indicate that here to bypass the “Comments to the Author” section, enter your conflict of interest statement in the “Confidential to Editor” section, and submit your "Accept" recommendation.

Reviewer #1: All comments have been addressed

Reviewer #3: All comments have been addressed

2. Does this manuscript meet PLOS Global Public Health’s publication criteria? Is the manuscript technically sound, and do the data support the conclusions? The manuscript must describe methodologically and ethically rigorous research with conclusions that are appropriately drawn based on the data presented.

Reviewer #1: Yes

Reviewer #3: Yes

3. Has the statistical analysis been performed appropriately and rigorously?

Reviewer #1: N/A

Reviewer #3: N/A

4. Have the authors made all data underlying the findings in their manuscript fully available (please refer to the Data Availability Statement at the start of the manuscript PDF file)?

Reviewer #1: Yes

Reviewer #3: Yes

5. Is the manuscript presented in an intelligible fashion and written in standard English?

Reviewer #1: Yes

Reviewer #3: Yes

6. Review Comments to the Author

Reviewer #1: (No Response)

Reviewer #3: Thank you for addressing the queries raised during the review process. The revisions have enhanced the clarity and coherence of the article, especially in the methodology and discussion/recommendation sections.

Congratulations on producing a fine article!

7. PLOS authors have the option to publish the peer review history of their article (what does this mean?). If published, this will include your full peer review and any attached files.

**Do you want your identity to be public for this peer review?** For information about this choice, including consent withdrawal, please see our Privacy Policy.

Reviewer #1: No

Reviewer #3: No
